# *Syzygium aromaticum* Extract Mitigates Doxorubicin-Induced Hepatotoxicity in Male Rats

**DOI:** 10.3390/ijms252312541

**Published:** 2024-11-22

**Authors:** Alaa Muqbil Alsirhani, Amal S. Abu-Almakarem, Maha Abdullah Alwaili, Salwa Aljohani, Ibtisam Alali, Aljazi Abdullah AlRashidi, Najlaa Yousef Abuzinadah, Sahar Abdulrahman Alkhodair, Maysa A. Mobasher, Tahiyat Alothaim, Thamir M. Eid, Karim Samy El-Said

**Affiliations:** 1Department of Chemistry, College of Science, Jouf University, Sakaka 72341, Saudi Arabia; amassaf@ju.edu.sa (A.M.A.); ikalali@ju.edu.sa (I.A.); 2Department of Basic Medical Sciences, Faculty of Applied Medical Sciences, Al-Baha University, Al Baha 65431, Saudi Arabia; amala2050@yahoo.com; 3Department of Biology, College of Science, Princess Nourah bint Abdulrahman University, Riyadh 11671, Saudi Arabia; maalwaele@pnu.edu.sa; 4Chemistry Department, Faculty of Science, Taibah University, Yanbu Branch, Yanbu 46423, Saudi Arabia; saajohani@taibahu.edu.sa; 5Chemistry Department, Faculty of Science, University of Ha’il, Ha’il 81451, Saudi Arabia; a.alrashedy@uoh.edu.sa; 6Department of Biological Science, College of Science, University of Jeddah, Jeddah 23714, Saudi Arabia; nyabuzinadah@uj.edu.sa; 7Department of Biochemistry, Faculty of Science, King Abdulaziz University, Jeddah 21589, Saudi Arabia; salkhodair@kau.edu.sa (S.A.A.); tmeid@kau.edu.sa (T.M.E.); 8Department of Pathology, Biochemistry Division, College of Medicine, Jouf University, Sakaka 72388, Saudi Arabia; mmobasher@ju.edu.sa; 9Department of Biology, College of Science, Qassim University, Buraydah 51452, Saudi Arabia; t.alothaim@qu.edu.sa; 10Biochemistry Division, Chemistry Department, Faculty of Science, Tanta University, Tanta 31527, Egypt

**Keywords:** *Syzygium aromaticum*, antioxidants, doxorubicin, ferroptosis, hepatotoxicity

## Abstract

Doxorubicin (DOX), an anticancer drug, is used to treat several types of tumors, but it has detrimental side effects that restrict its therapeutic efficacy. One is the iron-dependent form of ferroptosis, which is characterized by elevated ROS production and iron overload. *Syzygium aromaticum* has a diverse range of biological and pharmaceutical actions due to their antioxidant properties. This study investigated the effect of *S. aromaticum* extract (SAE) on hepatotoxicity caused by DOX in rats. Phytochemical analysis was performed to assess compounds in SAE. The ADMETlab 2.0 web server was used to predict the pharmacokinetic properties of the most active components of SAE when DOX was injected into rats. Molecular docking studies were performed using AutoDock Vina. Forty male Sprague Dawley rats were divided into four groups of ten rats each (G1 was a negative control group, G2 was given 1/10 of SAE LD_50_ by oral gavage (340 mg/kg), G3 was given 4 mg/kg of DOX intraperitoneally (i.p.) once a week for a month, and G4 was administered DOX as in G3 and SAE as in G2). After a month, biochemical and histopathological investigations were performed. Rats given SAE had promising levels of phytochemicals, which could significantly ameliorate DOX-induced hepatotoxicity by restoring biochemical alterations, mitigating ferroptosis, and upregulating the NRF-2–SLC7A-11–GPX-4 signaling pathway. These findings suggest that SAE could potentially alleviate DOX-induced hepatotoxicity in rats.

## 1. Introduction

Doxorubicin (DOX) is a chemotherapeutic anthracycline that is used to treat various types of cancer; however, it has some side effects that limit its efficacy, such as encouraging the accumulation of hazardous intermediates that have been linked to the production of liver injury [1]. Several studies have reported that DOX treatment negatively affected liver tissues and was associated with distinct biochemical and histopathological alterations [2,3]. Oxidative stress, apoptosis, and inflammation in the liver tissues were the main mechanisms with which DOX induced hepatotoxicity [4]. Ferroptosis is one of the recently identified molecular pathways associated with the pathophysiology of several disorders [5]. It is a major factor in the development of liver injury and is a possible pharmaceutical target [6]. The main biochemical features of ferroptosis include intracellular iron overload, lipid peroxidation, and antioxidant imbalance [7].

Nuclear factor E2-related factor 2 (NRF-2) regulates the core antioxidant response that controls essential defensive mechanisms in cells and protects against cellular damage [8]. A recent investigation reported that the regulation of solute carrier family 7, member 11 (SLC7A-11) had a role in protecting cells against ferroptotic death, and that this aided in the production of glutathione (GSH) [9]. Other proteins that are closely linked to ferroptosis are ferritin heavy polypeptide 1 (FTH-1) and glutathione peroxidase 4 (GPX-4) [10]. A previous study demonstrated the molecular mechanisms and therapeutic targeting of ferroptosis in DOX-induced organ toxicity [11]. Furthermore, it has been reported that regulating the NRF2–GPx4 axis with natural compounds inhibits ferroptosis in CCl_4_-induced liver injury in mice [12]. Ferritinophagy was mediated by DOX, which releases significant amounts of redox-active iron. The biochemical process leading to ferroptosis is Fenton’s reaction, which causes an excessive build up of ferrous iron (Fe^2+^) in the cells [13]. Therefore, the pharmacological targeting of ferroptosis could be a novel interventional modality for inhibiting DOX-induced toxicity in patients receiving DOX chemotherapy.

Natural antioxidants connect to ferrous iron (Fe^2+^), thereby preventing the generation of reactive intermediates that harm cells [14,15]. There have been reports on the preventive and therapeutic uses of herbal constituents that act against DOX-induced hepatotoxicity [16]. A previous report indicated a beneficial role of naringin in preventing DOX-induced liver damage; it did this by upregulating sirtuin 1 (SIRT1), which lowered oxidative stress, inflammation, and apoptosis [17]. Another study reported on the potential ameliorative effects of beetroot ethanolic extract against DOX-induced hepatic damage in rats [18]. *Allium cepa* extract indicated against liver damage caused by DOX in rats [19]. *Syzygium aromaticum* (clove), belonging to the Myrtaceae family, is cultivated in Brazil, Egypt, Madagascar, and Morocco [20]. It has several biological, biomedical, gastronomic, and traditional uses in medicine because of its high concentration of bioactive ingredients, including eugenol and flavonoids [21]. Some of its numerous pharmaceutical activities include antimicrobial, antiparasitic, antioxidant, anti-inflammatory, and anesthetic effects [22]. Pourlak et al. (2020) reported that *S. aromaticum* extract (SAE) treatment led to a significant mitigation of hepatic cell damage and oxidative stress in rats [23]. Furthermore, SAE significantly reduced levels of liver function enzymes and inflammatory cytokines in rats intoxicated with CCl_4_ [24]. It has been reported that SAE markedly improved the antioxidant status of Wistar rats intoxicated with artesunate [25]. The ameliorative efficacy of SAE has been reported to be associated with behavioral changes in lead-induced neurotoxicity in mice [26]. This novel study investigated the efficacy of SAE in ameliorating DOX-promoted liver damage in male rats.

## 2. Results

### 2.1. Phytochemical Components and GC-MS Analysis of SAE

The results show that the *S. aromaticum* flower buds (SAFBs) yielded an adequate extract amount (14%). The total phenolic and flavonoid contents of ASF were represented by 31.86 ± 2.43 mg GAE/g DW and 18.65 ± 1.58 mg QE/g DW, respectively. The TAC of SAE was 69.37 ± 3.79 mg AAE/g DW. The saponin and anthocyanin levels were 365 ± 4.37 mg/g DW and 7.89 ± 0.45 mg ECG/g DW, respectively. The DPPH scavenging activity was 82.64% ± 3.95. The IC_50_ was 6.05 ± 0.86 mg/mL (Table 1).

GC-MS analysis showed that SAE contains promising phytochemicals, the most abundant being ethyl-α-D-glucopyranoside, eugenol, caryophyllene, and humulene. The retention times (RTs) were 7.64, 12.25, 15.82, and 16.54 min, respectively. The peak area percentages (PA%) were 2.27%, 64.17%, 18.07%, and 3.83%, respectively (Figure 1 and Table 2).

### 2.2. In Silico ADMET Analysis

ADMET screening provided a comprehensive comparison of the pharmacokinetic and toxicological properties of DOX and revealed that eugenol was the most abundant compound in SAE. Eugenol had better oral bioavailability (F20% and F30%) and blood–brain barrier penetration (BBB) than DOX. DOX has a high likelihood of being a P-glycoprotein substrate, which may affect its cellular accumulation and efficacy. Furthermore, eugenol and 10-Heptadecen-8-ynoic acid, methyl ester showed plasma-protein-binding affinities with lower fractions unbound (Fu) than DOX, indicating its active transport through the bloodstream. In addition, eugenol demonstrated a lower probability of various toxic endpoints. The toxicity prediction results for eugenol and DOX further highlight the differences in their safety profiles. Eugenol and 10-Heptadecen-8-ynoic acid, methyl ester are predicted to be inactive against most organ toxicities. In contrast, DOX was predicted to be active versus multiple organ toxicities, including cardiotoxicity and hepatotoxicity (Figure 2 and Table 3). The bioconcentration factors (BCFs) of eugenol and 10-Heptadecen-8-ynoic acid (1.203) and methyl ester (2.424) were higher than those of DOX (0.567), which suggests their high degree of uptake from the dissolved phase. The quantitative estimate of drug-likeness (QED) showed that eugenol and 10-Heptadecen-8-ynoic acid, methyl ester were closer to 1, indicating that they were more drug-like than DOX. The Lipinski filter is used to filter out any drug at the absorption or permeation level (an ideal drug has a molecular weight of less than 500 g/mol). The molecular weight of eugenol was 164.08 g/mol and that of 10-Heptadecen-8-ynoic acid, methyl ester was 278.22 g/mol, so they were acceptable, while that of DOX (543.17 g/mol) was not. The Veber (GSK) rule defines drug-likeness constraints as a rotatable bond count ≤ 10 and a topological polar surface area (TPSA) ≤ 140. Our ADMET screening showed that eugenol and 10-Heptadecen-8-ynoic acid, methyl ester followed the rule and were acceptable by GSK standards, while DOX was not (Table 3). The AMES test indicated that toxicity, hepatotoxicity, and skin sensitization were not found in the therapeutic compounds compared with DOX. The overall analysis demonstrated that eugenol was more biodegradable and bioavailable and has physicochemical, molecular, and ADMET properties that lie between the upper and lower predicted values (see Figure 2 and Table 3).

### 2.3. Bioactive Compound (Eugenol) Interactions with Target Proteins (NRF-2, SLC7A-11, and GPX-4) as Shown by Molecular Docking Analysis

The molecular docking results presented in Table 4 show the ∆G and binding affinities of the compounds detected in SAE and DOX for the key proteins involved in cellular protection and the oxidative stress response (NRF-2, SLC7A-11, and GPX-4). DOX consistently demonstrated stronger binding affinities across the three targets NRF-2 (−7.8), SLC7A-11 (−8.4), and GPX-4 (−6.7) compared with other compounds. 10-heptadecen-8-ynoic acid, methyl ester and eugenol showed lower binding affinities (Table 4). This suggests that DOX may have a greater inhibitory effect on these proteins, and this could contribute to its potential toxicity. DOX formed more hydrogen bonds and had a greater variety of interactions than 10-heptadecen-8-ynoic acid, methyl ester (NRF-2 (−4.3), SLC7A-11 (−5.3), and GPX-4 (−3.9)) and eugenol (NRF-2 (−4.7), SLC7A-11 (−6.0), and GPX-4 (−4.3)). This extensive binding pattern of DOX could potentially disrupt GPX-4 function significantly, which might compromise the cells’ antioxidant defenses. The interactions with NRF-2 showed a similar trend, with DOX forming multiple hydrogen bonds and hydrophobic interactions, whereas eugenol primarily engaged in hydrophobic interactions. NRF-2 is a crucial regulator of the cellular response to oxidative stress, and the stronger binding of DOX might interfere with its protective functions to a greater extent than eugenol. For SLC7A-11, both compounds showed fewer interactions than the other targets, but DOX still demonstrated stronger binding. SLC7A-11 is involved in glutathione synthesis, an important cellular antioxidant. The different binding patterns suggest that DOX might have a greater impact on this cytoprotective pathway (Figure 3, Figure 4 and Figure 5).

### 2.4. The Oral LD_50_ of SAE, Body Weight, and Liver Weight Changes After Treatment with DOX–SAE

The LD_50_ of SAE after oral administration was determined for dosages of 1000 to 5000 mg/kg in different groups of rats after 24 h of treatment. The rats showed no stereotypical toxic symptoms except at 3400 mg/kg. The probit analysis showed that the oral LD_50_ of SAE was 3400 mg/kg.

The DOX-injected group showed a significant decrease (*p* < 0.05) in % b. wt. change (16.42%) compared with that in the control groups. Treatment of DOX-exposed rats with SAE led to a significant improvement (*p* < 0.05) in % b. wt. changes to 28.75% compared with that in the DOX-treated group (Figure 6A). Among the experimental groups under the study, the relative liver weight of the DOX-injected group increased (see Figure 6B).

### 2.5. Treatment with SAE Restored Hepatic Function Markers in DOX-Injected Rats

In the DOX-injected group, levels of liver transaminases (ALT and AST) were significantly increased (*p* < 0.05) (to 71.23 ± 1.78 and 103.67 ± 2.78 U/L, respectively). However, the treatment of DOX-injected rats with SAE led to a considerable reduction in the ALT and AST levels (to 38.93 ± 0.85 and 60.78 ± 2.56 U/L, respectively) (Table 5). Furthermore, the results show that DOX-exposed rats showed a significant increase (*p* < 0.05) in ALP level, representing 486.98 ± 6.34 U/L compared with the control groups. However, treatment with SAE demonstrated a significant decrease in ALP serum level (317.92 ± 5.79 U/L) compared with the DOX-injected group alone. The total and direct bilirubin levels in the DOX-injected group increased significantly (*p* < 0.05), but SAE therapy decreased these levels (see Table 5).

### 2.6. Treatment with SAE Alleviated Hepatic Oxidative Stress Induced by DOX in Rats

The DOX-treated rats showed a significant increase (*p* < 0.05) in their hepatic MDA levels (to 0.542 ± 0.025 nmol/mg protein) versus the negative control (0.258 ± 0.019 nmol/mg protein) and SAE control groups (0.231 ± 0.014 nmol/mg protein) (Figure 7A). Hepatic SOD and CAT activities significantly decreased (*p* < 0.05) in the hepatic tissues of DOX-injected rats when compared with the control groups; however, concomitant treatment with DOX and SAE led to a significant restoration of SOD and CAT activities (Figure 7B,C). Moreover, the GSH level was significantly decreased in the DOX-injected group (by −2.2 fold) compared with the control groups (Figure 7B). The group that was treated with DOX–SAE showed a significant increase in the GSH level as compared with the DOX-administered group alone (25.15 ± 1.14 vs. 14.12 ± 0.48 mg/mg protein) (Figure 7D).

### 2.7. SAE Treatment Modulated Ferroptosis-Related Proteins and Their Gene Expressions in DOX-Exposed Rats

Figure 8A demonstrates that, compared with the hepatic iron level in the negative control and SAE control groups (2.1 ± 0.16 and 1.8 ± 0.14 µmol/g protein, respectively), the DOX-injected group had a significant increase in hepatic iron levels (4.72 ± 0.31, *p* < 0.01). The group treated with DOX–SAE had a significant reduction (*p* < 0.01) in the hepatic iron level. To investigate the effect of DOX–SAE treatment on specific ferroptosis-related parameters in rats, the levels of NRF-2, SLC7A-11, and GPX-4 were determined in the rats’ livers. There were significant reductions (*p* < 0.01) in these proteins in the DOX-treated group compared with those in the control groups. However, treatment with SAE led to a significant restoration of the previously mentioned ferroptosis-related parameters in hepatic rats when compared with the DOX-injected group alone (see Figure 8).

The total protein levels in the hepatic tissues were measured using the method of Lowry et al. (1951) with bovine serum albumin as a standard. The DOX-exposed group showed a significant decrease (*p* < 0.05) in the hepatic protein concentration, whereas the combination of DOX and SAE treatments resulted in a significant increase in hepatic protein concentrations. Furthermore, compared with the negative control and SAE control groups (28.2 ± 0.98 and 31.1 ± 1.10 ng/mg protein, respectively), the protein level of FTH-1 in the DOX-injected group was considerably lower in the rats’ hepatocytes (13.3 ± 0.79 ng/mg protein, *p* < 0.05). Meanwhile, the DOX–SAE-treated group showed a significant increase in the FTH-1 level (21.2 ± 0.95 ng/mg protein, *p* < 0.05) compared with the DOX-injected group alone. In contrast, ACSL-4 and NCOA-4 protein levels were significantly increased in the DOX-treated group versus the control groups. However, these increases in ACSL-4 and NCOA-4 levels were significantly attenuated (*p* < 0.05) in the setting of DOX–SAE treatment (Figure 9).

Molecular analysis of ferroptosis-related genes by the RT-PCR method using glyceraldehyde-3-phosphate dehydrogenase (*GAPDH*) as a housekeeping gene showed that DOX injection in rats led to a significant downregulation (*p* < 0.001) in the relative mRNA expression levels of the *NRF2*, *SLC7A11*, *GPX4*, and *FTH1* genes (by 2.5, 1.6, 1.4, and 4-fold, respectively). In contrast, the fold changes in the *NCOA4* gene were significantly upregulated (*p* < 0.001) in the DOX-challenged group (by 3.7 fold) (Table 6). Interestingly, concomitant treatment with DOX and SAE led to a significant upregulation of *NRF2*, *SLC7A11*, *GPX4*, and *FTH1* along with a significant downregulation of *NCOA4* (see Table 6).

### 2.8. Treatment with SAE Reduced Hepatic Inflammatory Cytokines in Rats Given DOX

The results show that, in the group injected with DOX alone, the levels of inflammatory biomarkers, including IL-6, IL-1β, TNF-α, NF-κB, and COX-2, were significantly elevated (*p* < 0.001) compared with the control groups. However, these inflammatory cytokines were significantly decreased (*p* < 0.01) in the DOX–SAE-administered group (Table 7).

### 2.9. Treatment with SAE Improved Hepatic Histopathological Changes Induced by DOX in Rats

In H&E-stained liver sections, histopathological analysis revealed a normal hepatocyte architecture and a central hepatic vein with centrally located nuclei in the liver sections of the negative control group and the group given SAE; their pathological scores were 0.12 ± 0.09 and 0.10 ± 0.11, respectively (Table 8, Figure 10A,B). The liver sections of DOX-treated rats showed extensive hepatocyte degeneration, with an acutely dilated central vein, cellular swelling, and nuclear changes. The semi-quantitative analysis showed high pathological scores (3.50 ± 0.21; Table 8, Figure 10C). The liver section of the group given DOX–SAE displayed a significant improvement in the hepatic architecture and less congestion (see Table 8 and Figure 10D).

## 3. Discussion

The anthracycline DOX is one of the most commonly used systemic anti-neoplastic drugs. Its clinical efficacy is hampered, however, by the toxicities it induces. Hepatotoxicity is a frequent, serious adverse consequence caused by oxidative stress [2]. To date, there have been no specific and efficient treatment options available for the hepatotoxicity promoted by DOX. Therefore, it is imperative to evaluate natural constituents that might raise the therapeutic index of DOX while lowering its adverse effects [4]. Various natural constituents have been reported to exert certain curative effects on liver injuries by targeting ferroptosis [27]. Clove is one of the spices used in various dishes and is interestingly utilized for several therapeutic uses in traditional medicine related to its bioactive chemical constituents [28]. NRE-2 plays a key role in the process of ferroptosis by regulating a series of proteins, including FTH-1, GPX-4, SLC7A-11, and HO-1 [29]. Interestingly, some plant extracts could be effectively used as therapeutic targets to alleviate liver injuries by regulating NRF-2 or downstream effector proteins inhibiting ferroptosis [5,30]. Recently, a noted increase in the expression of GPX-4 and SLC7A-11 in liver tissues suggested that natural products may inhibit ferroptosis [31]. The current investigation addressed the ameliorative effect of SAE treatment on DOX-induced hepatotoxicity in male rats.

The results obtained from the current study show that, as demonstrated in Table 1, the flower buds of *S. aromaticum* yielded an adequate amount of extract and contained promising phytochemical constituents, including phenols, flavonoids, saponins, and anthocyanins, and have high percentages of DPPH scavenging activity [20,21,28,32]. Furthermore, GC-MS analysis of SAE revealed the presence of pharmacologically important phytochemical compounds, including ethyl-α-D-glucopyranoside, eugenol, caryophyllene, and humulene (see Table 2 and Figure 1). Consistently with our investigation, previous reports demonstrated that SAE contains bioactive compounds and that eugenol was the richest compound found when different extraction methods were used [28,33].

In the context of cytoprotection and cytotoxicity in hepatic cells, the present results suggest that eugenols may offer a more favorable safety profile, while the stronger interactions of DOX with antioxidants and stress-response proteins might contribute to its hepatotoxic effects [4]. By examining the types of interactions and their distances, we can infer the strength and specificity of binding, which can correlate with biological activity. For GPX-4, DOX forms multiple conventional hydrogen bonds with distances of 2.00 to 2.81 Å. These short hydrogen bonds indicate strong, specific interactions that are likely to significantly influence the functioning of GPX-4 [34]. Additionally, DOX was engaged in electrostatic and hydrophobic interactions, contributing to a stable binding pose. In contrast, eugenol’s interaction with GPX-4 was characterized by a single conventional hydrogen bond (2.24 Å) and several hydrophobic interactions. The fewer and generally weaker interactions suggest that eugenol may have had a more moderate effect on GPX-4 activity than DOX. With NRF-2, DOX demonstrated multiple conventional hydrogen bonds with distances between 2.05 and 2.55 Å, indicating strong binding. The presence of several hydrophobic interactions further stabilizes the binding [35]. Eugenol’s interaction with NRF-2 is limited to hydrophobic interactions, with no hydrogen bonds observed. This suggests a weaker and potentially less specific binding to NRF-2 compared with DOX, which may have led to a reduced impact on NRF2-mediated pathways [36]. However, for SLC7A-11, eugenol shows two conventional hydrogen bonds with distances of 2.13 and 2.97 Å, along with hydrophobic interactions. This binding pattern suggests a moderate interaction with SLC7A-11. The interaction of DOX with SLC7A-11 is characterized by a single conventional hydrogen bond (2.07 Å) and two pi–alkyl interactions. The fewer interactions observed for both compounds with SLC7A-11 than the other targets may indicate that this protein is less significantly affected by either compound. Specifically, the conventional hydrogen bonds, particularly those under 3 Å, were crucial for specific and strong binding [37]. DOX consistently formed more of these bonds across all targets, suggesting it may more potently modulate the activities of these proteins. Therefore, hydrophobic interactions played a role in the binding stability. While these interactions were generally weaker than hydrogen bonds, their cumulative effect could significantly impact the binding affinity. DOX formed some electrostatic interactions, particularly with GPX-4, which could enhance the binding specificity and strength. In addition, the interaction distances for the shorter interaction distances observed (mostly between 2 and 4 Å) indicate strong binding for both compounds, with DOX generally showing shorter distances and potentially stronger interactions. The diversity of the interactions, especially DOX, consistently demonstrates a more diverse range of interaction types across all targets, suggesting a more complex and potentially more disruptive binding mode [38]. These interaction patterns reveal that, while both compounds could bind to these cytoprotective proteins, DOX’s bindings are generally stronger and more extensive. This could explain its higher potency and its greater potential for disrupting normal cellular functions, which could lead to side effects [39].

At a dose of 1000 mg/kg body weight per day, cloves have been reported to have no toxic effects on Wistar rats [40]. The present study found that the oral LD_50_ of SAE was 3400 mg/kg after 24 h of treatment in rats. A previous study on oral toxicity revealed that the LD_50_ of *S. aromaticum* essential oil was approximately 4500 mg/kg, demonstrating the safety of SAE oral administration [41]. Treatment with DOX–SAE led to a significant improvement in the % b. wt. and relative liver weight changes (Figure 6). This finding was consistent with previous research showing the impact of natural plant constituents on DOX-induced body weight loss in experimental animals [4,42]. Our findings demonstrate that SAE treatment could reduce elevated ALT, AST, and ALP activities as well as the total and direct bilirubin levels compared with those activities in DOX-injected rats, indicating the therapeutic properties of SAE against DOX-promoted liver damage. These results are in line with those of numerous studies highlighting the hepatoprotective potential of plant extracts to limit DOX-induced liver injuries in experimental animals [4,24,43,44]. Additionally, data from our study reveal that DOX injection in rats resulted in a significant increase in their hepatic MDA levels and a significant decrease in their hepatic SOD, CAT, and GSH levels (see Figure 7). This could be attributed to the increased production of reactive free radicals by the DOX-exposed liver tissues, and, hence, the peroxidation of the lipid membrane and increased MDA generation [4,45]. Increased lipid peroxidation damages liver tissue and impairs antioxidant defense mechanisms, increasing the risk that metabolites and enzymes might leak into the bloodstream. Co-treatment with DOX–SAE significantly restored these oxidants/antioxidants’ biomarkers. These data indicate the promising role of SAE in enhancing the impact of antioxidants/hemostasis during DOX treatment in rats. The increased GSH levels resulting from SAE treatment helped to eliminate active DOX metabolites and scavenge free radicals involved in lipid oxidation, reversing hepatic damage. Similar effects of SAE treatment have been reported in experimental animals, which confirmed the hepatoprotective action of cloves [23,24,46,47].

Ferroptosis is one of the types of cell death that have been identified in recent years. It is associated with lipid peroxidation and significant iron build up and plays a role in several pathophysiological processes [48]. The results show that DOX injection increased the hepatic iron content in rats compared with that in the control group, suggesting the increased storage of hepatic iron, which in turn induced the generation of ROS and lipid peroxide, thereby leading to ferroptosis [49]. GPX-4 protects cells against lipid peroxidation, and the depletion of GPX-4 is one of the main initiators of ferroptosis [50]. For the first time, our investigation found that DOX decreased the hepatic levels of GSH and GPX-4 in rats in a NRF2-dependent manner, which could promote ferroptosis. Treatment with SAE led to a significant increase in hepatic GPX-4 expression, suggesting the effect of SAE in reversing ferroptosis induced by DOX injection (see Table 6 and Figure 8). SLC7A-11 can maintain intracellular GSH levels, and its reduction will inhibit GPX-4 expression. The SLC7A-11–GPX-4 pathway is an important regulatory pathway in ferroptosis, and the inhibition of the SLC7A-11–GPX-4 pathway could indicate the occurrence of ferroptosis [51]. In the current study, the hepatic SLC7A-11 levels were decreased after DOX poisoning; this finding supports the conclusion that the inhibition of the SLC7A-11–GPX-4 pathway may have negative effects on DOX intoxication in rats. In contrast, SAE treatment led to a significant increase in the protein level of SLC7A-11.

Interestingly, FTH-1, a key subunit of ferritin, is involved in signaling pathways in several diseases and in maintaining the cellular iron balance during ferroptosis. The process is mediated by NCOA-4, which selectively binds FTH-1, resulting in iron release [52]. High iron concentrations inhibit the binding of FTH-1 to NCOA-4 and enhance the degradation of NCOA-4, leading to the inhibition of ferritin degradation and ferroptosis [53]. In this study, the hepatic FTH-1 expression level was significantly downregulated in the DOX-injected group, while the NCOA-4 expression level was significantly improved. However, concomitant treatment with DOX–SAE significantly restored FTH-1 and NCOA-4 levels in the liver tissues of treated rats. These data suggest that SAE could be an FTH-1 inducer, a potent ferroptosis inhibitor that could be used as a pharmacological target for inhibiting ferroptosis induced by DOX. The current investigation reports for the first time a new mechanism of action of SAE that could ameliorate DOX-induced hepatotoxicity by targeting ferroptosis in rats. Additionally, DOX causes inflammation, as seen in the increased production of several proinflammatory cytokines in liver tissues [54]. Consequently, limiting inflammation might be a useful tactic to stop DOX hepatotoxicity. The present results show that the hepatic levels of inflammatory mediators, including IL-6, IL-1β, TNF-α, NF-κB, and COX-2, were significantly increased following DOX injection in rats. The increased level of these pro-inflammatory mediators in sequence leads to more hepatic injuries due to DOX exposure. However, these inflammatory cytokines were significantly restored in the DOX–SAE-treated group, and this finding suggests that the immunoregulatory properties of SAE acted against the inflammatory responses that were promoted by DOX injection in rats (see Table 7). These results may underline the mechanism of the anti-inflammatory effects exhibited by SAE. These findings are in accordance with previous studies that demonstrated the anti-inflammatory role of plant extracts against DOX-induced hepatic inflammation in experimental animals [4,55,56]. A previous study reported that Mokko lactone attenuated DOX-induced hepatotoxicity in rats through suppressing hepatic inflammatory infiltration [43]. Furthermore, Sandamali et al. (2022) reported that *Nauclea orientalis* (L.) bark extract protected rats from DOX-induced inflammation [57]. Cytochrome enzymes convert DOX into doxorubicinol and other hazardous byproducts that can damage the liver [43]. Histopathological changes affecting liver tissues, including hepatocyte vacuolation, have been linked to DOX treatment, according to a prior publication supporting the previous biochemical results [3]. In concordance with previous reports, our histopathological investigations of the liver section of DOX-injected rats showed extensive hepatocyte degeneration with a dilated central vein. The liver section of the DOX–SAE-treated group represented a significant improvement in the hepatic architecture (see Figure 10). These results suggest the beneficial role of SAE treatment in restoring hepatic cell damage that was promoted by DOX injection [3,4,43,44,56]. A previous study demonstrated that an *S. aromaticum* fraction rich in eugenol reversed biochemical and histopathological alterations in liver cirrhosis and suppressed hepatic cell injury [46].

## 4. Materials and Methods

### 4.1. Chemicals

Ethyl alcohol, ferric chloride (FeCl_3_), aluminum chloride (AlCl_3_), sodium nitroprusside, trichloroacetic acid (TCA), potassium ferricyanide (K_3_Fe_3_(CN)_6_), and gallic acid were purchased from Algomhoria Co., 23 El Sawah St., El Amiriya, Cairo, Egypt. Butyl hydroxy toluene (BHT) and 2,2-diphenyl-2-picrylhydrazyl (DPPH) (catalog no. 300267) were procured from Merck Co., Mumbai 400079, Maharashtra, India. Doxorubicin hydrochloride (catalog no. D5220, 98–102% HPLC) was purchased from Sigma-Aldrich (Oakville, ON L6H 6J8, Canada).

### 4.2. Collection and Preparation of Plant Materials

*S. aromaticum* flower buds were purchased from the Carrefour Market in Tanta, Egypt in November 2023. The plant was authenticated by a specialist and complied with the institutional guidelines. The chopped buds were ground into a powder, 50 g of the powdered flowers in 500 mL of ethanol (70%) were filtered, and *S. aromaticum* extract (SAE) was obtained [58].

### 4.3. Phytochemicals Analysis of SAE

The phytochemicals and DPPH radical scavenging capability were evaluated in the SAE. Butyl hydroxytoluene (BHT) was used as a standard antioxidant [59,60,61,62].

### 4.4. Gas Chromatography and Mass Spectrometry (GC-MS) Profiling of SAE

A Trace GC 1310-ISQ mass spectrometer (Thermo Scientific, Austin, TX, USA) was used to identify the phytochemicals in the SAE. The components were detected, and their mass spectral and retention periods were matched to those found in the mass spectral databases of WILEY 09 and NIST 11 [63].

### 4.5. Molecular Docking Analysis

The SMILES codes of the compounds doxorubicin and eugenol were used as the input for the ADMETlab 2.0 web server to predict their pharmacokinetic properties, including the absorption, distribution, metabolism, excretion, and toxicity (ADMET) [64]. The protein structures for NRF-2 (Uniprot ID: O54968), SLC7A-11 (Uniprot ID: D4ADU2), and GPX-4 (Uniprot ID: P36970) were collected from the UniProt database, and models were generated using the AlphaFold server [65]. The active sites of all proteins were predicted using the CB-Dock2 server, which employs deep learning algorithms to identify potential binding pockets and catalytic sites. The prepared protein structures were further processed using AutoDock Tools 1.5.7. The active constituents obtained from the ligand molecules were retrieved from the PubChem database in their respective SDF formats. These ligands were then minimized using the Avogadro 1.2.0 software, employing the Force Field algorithm (MMFF94) and the Conjugate Gradients algorithm [66,67]. The minimized ligand structures were converted to the PDBQT format compatible with AutoDock Vina. The search space for the docking simulations was defined based on the predicted active site regions obtained from the deep site server. The docking results were visualized and analyzed using BIOVIA Discovery Studio 2020 (San Diego, CA, USA).

### 4.6. Oral Median Lethal Dose (LD_50_) of SAE

The oral LD_50_ following SAE administration was evaluated in rats. Rats were gavaged with SAE (1–5 g/kg b. wt.) then observed to see if there were any signs of toxicity for 24 h. The LD_50_ value was determined by the probit analysis [68].

### 4.7. Animals and Experimental Design

Forty adult male Sprague Dawley rats (130–150 g, 5–6 weeks of age) were purchased from Helwan University, Egypt. Male rats were chosen in order to avoid the fluctuating hormone levels and the menstrual cycle that could make the data difficult to interpret and the results more variable. Our study was conducted in accordance with the Faculty of Science, Tanta University animal care committee (IACUC-SCI-TU-0321). Rats were divided into four groups (*n* = 10). G1 was a negative control group that was injected i.p. with saline daily, G2 was orally given 1/10 of SAE LD_50_ (340 mg/kg) daily for a month, G3 was injected with 4 mg/kg of DOX i.p. once a week for a month [69], and G4 received the same DOX injection as G3 and the same SAE administration as G2. Serum and liver tissues were collected for biochemical, molecular, and histopathological investigations.

### 4.8. Biochemical Analysis

Alanine aminotransferase (ALT) (catalog no. AL103145), aspartate aminotransferase (AST) (catalog no. AS106145), alkaline phosphatase (ALP) (catalog no. AP1020), total bilirubin (catalog no. BR1111), and direct bilirubin (catalog no. BR1112) were assessed using colorimetric kits (Spectrum Diagnostics, Egypt). Hepatic levels of malondialdehyde (MDA) (catalog no. MD2529), superoxide dismutase (SOD) (catalog no. SD2521), catalase (CAT) (catalog no. CA2517), and reduced glutathione (GSH) (catalog no. GR2511) were measured by using their respective kits (Biodiagnostic, Giza Governorate, Egypt). The protein concentration was measured by the method of Lowry et al. (1951) using bovine serum albumin (BSA) as a standard [70]. Iron levels in rats’ livers were detected using the iron assay kit (catalog no. ab83366). Hepatic nuclear factor erythroid 2-related factor 2 (NRF-2) (catalog no. MBS752046), solute carrier family 7, member 11 (SLC7A-11) (catalog no. MBS2705481), glutathione peroxidase (GPX-4) (catalog no. MBS934198), ferritin heavy chain-1 (FTH-1) (catalog no. MBS2886777), acyl-CoA synthetase long-chain family member 4 (ACSL-4) (catalog no. MBS9903690), and nuclear receptor coactivator 4 (NCOA-4) (catalog no. MBS7269622) were determined using the respective rat-specific ELISA kits from MyBioSource, Inc., San Diego, CA, USA. Rat-specific ELISA kits were used for the measurement of the inflammatory biomarkers, including interleukin-6 (IL-6) (catalog no. E-HSEL-R0004), interleukin-1β (IL-1β) (catalog no. E-EL-R0012), tumor necrosis factor-α (TNF-α) (catalog no. RAB0479), nuclear factor kappa-B (NF-κB) (catalog no. MBS453975), and cyclooxygenase-2 (COX-2) (catalog no. MBS266603) in the liver homogenates of the different groups.

### 4.9. Molecular Analysis

The mRNA expression of the *NRF2*, *SLC7A11*, *GPX4*, *FTH1*, and *NCOA4* genes was evaluated in liver tissues using *GAPDH* as an internal reference. The primers were prepared using the Primer-Blast program from NCBI (Table 9). The relative expression of target genes was estimated [71].

### 4.10. Histopathological Investigations

Liver tissues were sectioned at 5 μm, embedded in paraffin wax, washed in xylene, and then sliced and fixed in 10% buffered formalin. Hematoxylin and eosin (H&E) staining was applied to the sections, which were then viewed under an Olympus CX31 light microscope and captured on a digital camera (Olympus Camedia 5060, Tokyo, Japan) [72]. Hepatic damage was analyzed based on the severity and tissue damage percentage. A scale (0–4) was used, where 0 is normal tissue, 1 is <25% hepatic tissue damage, 2 is 26–50% hepatic tissue damage, 3 is 51–75% hepatic tissue damage, and 4 is >75% hepatic tissue damage [73].

### 4.11. Statistical Analysis

A one-way analysis of variance (ANOVA) was used to assess significant variations. The software GraphPad Prism, Inside Scientific Co., (San Diego, CA, USA), https://www.graphpad.com/ (accessed on 5 April 2024) was utilized for the evaluation of results. For multiple comparisons, Tukey’s test was applied, and statistical significance was established at *p* < 0.05.

## 5. Conclusions

SAE may be a highly effective protective agent against hepatotoxicity brought on by DOX therapy. It has this effect through its inhibition of oxidative stress and inflammation and its upregulation of the NRF-2–SLC7A-11–GPX-4 signaling pathway. Further preclinical and clinical studies should investigate the effects of SAE against other toxicities induced by DOX.

## Figures and Tables

**Figure 1 ijms-25-12541-f001:**
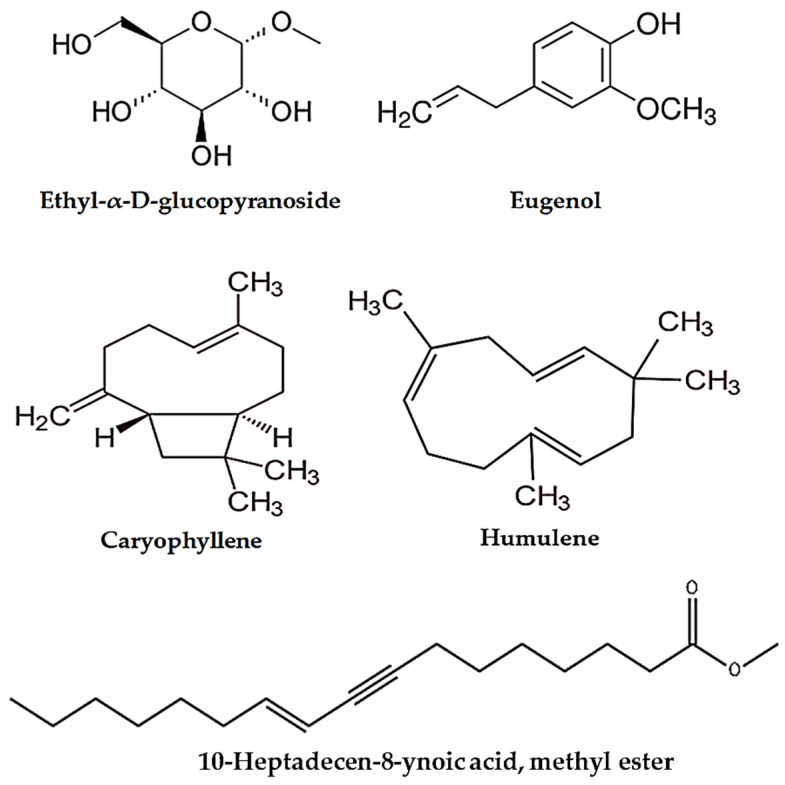
The abundant bioactive compounds in SAE.

**Figure 2 ijms-25-12541-f002:**
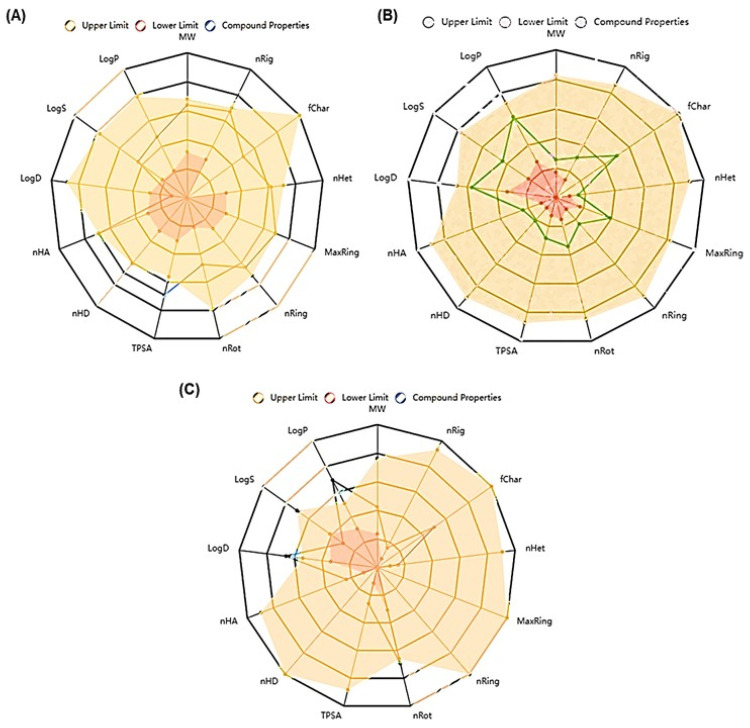
Radar chart for ADMET screening showing the upper, lower, and predicted values of various physicochemical and molecular properties of DOX (**A**), eugenol (**B**), and 10-Heptadecen-8-ynoic acid, methyl ester (**C**). MW, molecular weight; nRig, number of rigid bonds; fChar, formal charge; nHet, number of heteroatoms; MaxRing, number of atoms in the biggest ring; nRing, number of rings; nRot, number of rotatable bonds; TPSA, topological polar surface area; nHD, number of hydrogen bond donors; nHA, number of hydrogen bond acceptors; LogP, Log of the octanol–water partition coefficient; LogS, Log of the aqueous solubility; LogD, Log at physiological pH (7.4).

**Figure 3 ijms-25-12541-f003:**
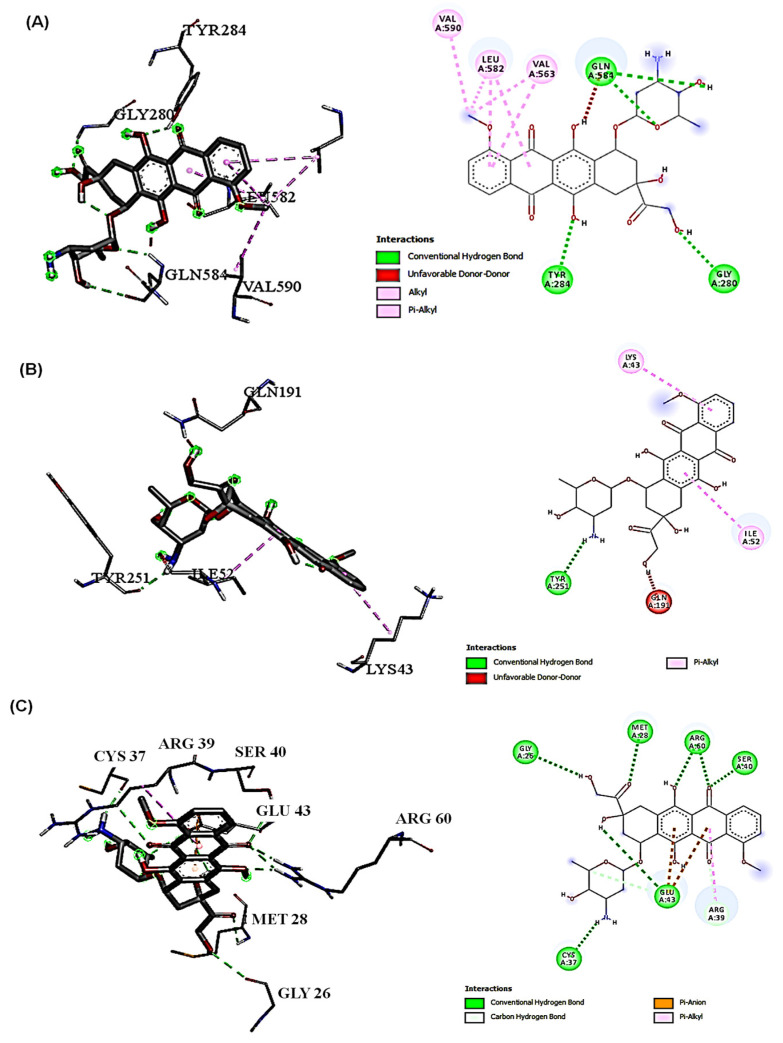
DOX interactions with three target proteins (NRF-2, SLC7A-11, and GPX-4) (2D and 3D). (**A**) DOX interaction with NRF-2, the conventional H-bond (GLY280, TYR284, GLN584, and GLN584), and the hydrophobic interactions (VAL563, LEU582, and VAL590); (**B**) DOX interaction with SLC7A-11, the conventional H-bond (TYR251), and the hydrophobic interactions (ILE52 and LYS43); (**C**) DOX interaction with GPX-4, the conventional H-bond (MET28, SER40, ARG60, GLU43, GLY26, CYS37, and LYS595), the carbon H-bond (ARG39 and GLU43), the Pi–Anion interaction (GLU43), and the Pi–Alkyl interaction (ARG39).

**Figure 4 ijms-25-12541-f004:**
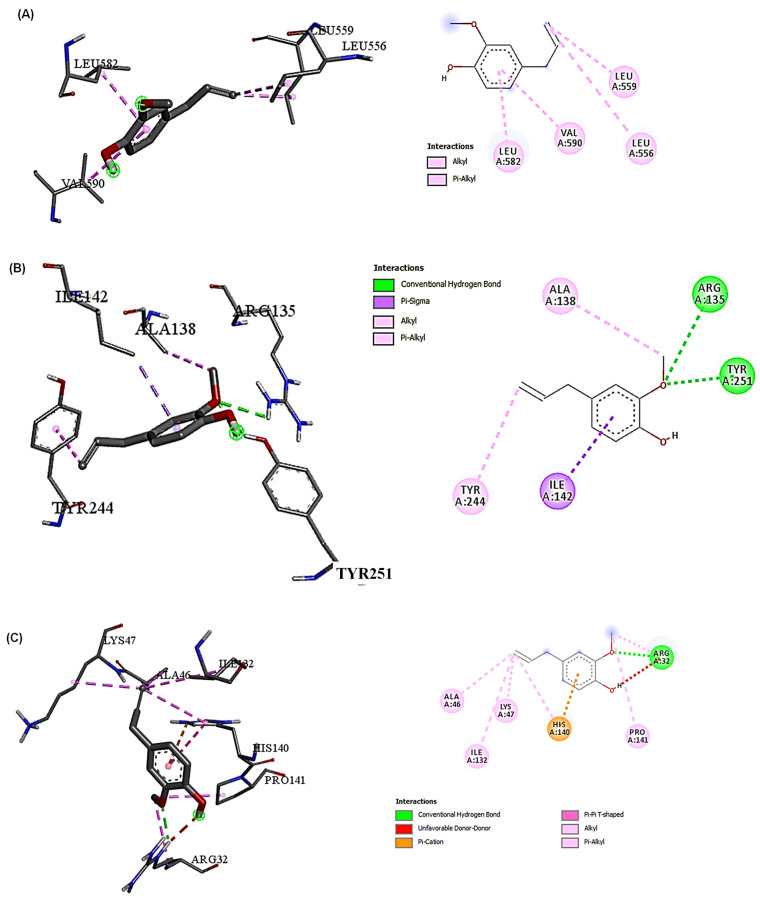
Eugenol interactions with three target proteins (NRF-2, SLC7A-11, and GPX-4) (2D and 3D). (**A**) Eugenol interaction with NRF-2 and the hydrophobic interactions (LEU556, LEU559, LEU582, and VAL590); (**B**) Eugenol interaction with SLC7A-11, the conventional H-bond (ARG135 and TYR251), and the hydrophobic interactions (ILE142, ALA138, and TYR244); (**C**) Eugenol interaction with GPX-4, the conventional H-bond (ARG32), and the hydrophobic interactions (HIS140, ALA46, ARG32, PRO141, LYS47, ILE132, and HIS140).

**Figure 5 ijms-25-12541-f005:**
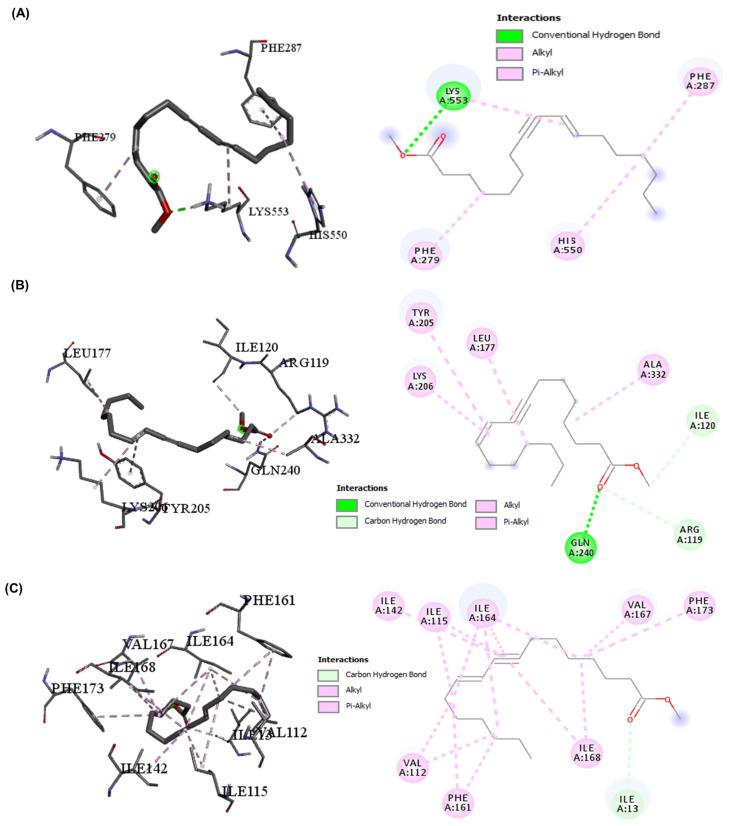
10-Heptadecen-8-ynoic acid, methyl ester interactions with three target proteins (NRF-2, SLC7A-11, and GPX-4) (2D and 3D). (**A**) The interaction with NRF-2, the conventional H-bond (LYS553), and the hydrophobic interactions (PHE279, PHE287, and HIS550); (**B**) The interaction with SLC7A-11, the conventional H-bond (GLN240), the carbon H-bond (ARG119 and ILE120), and the hydrophobic interactions (LYS206, ALA332, LEU177, and TYR205); (**C**) The interaction with GPX-4, the carbon H-bond (ILE13), and the hydrophobic interactions (VAL112, ILE115, ILE142, ILE164, VAL167, ILE168, PHE161, and PHE173).

**Figure 6 ijms-25-12541-f006:**
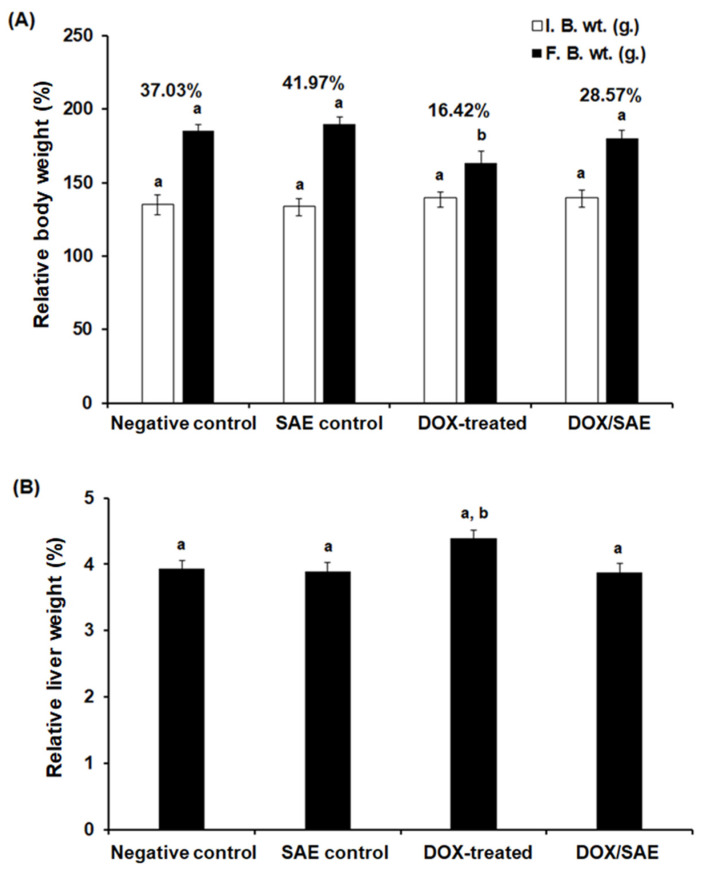
(**A**) The relative body weight percentages. (**B**) The relative liver weight in the different groups under study. I.B.W, initial body weight; F.B.W, final body weight; SAE, *Syzygium aromaticum* extract; DOX, Doxorubicin. Data are expressed as mean ± S.D., *n* = 10. Means that do not share a letter showed a significant difference (*p* < 0.05).

**Figure 7 ijms-25-12541-f007:**
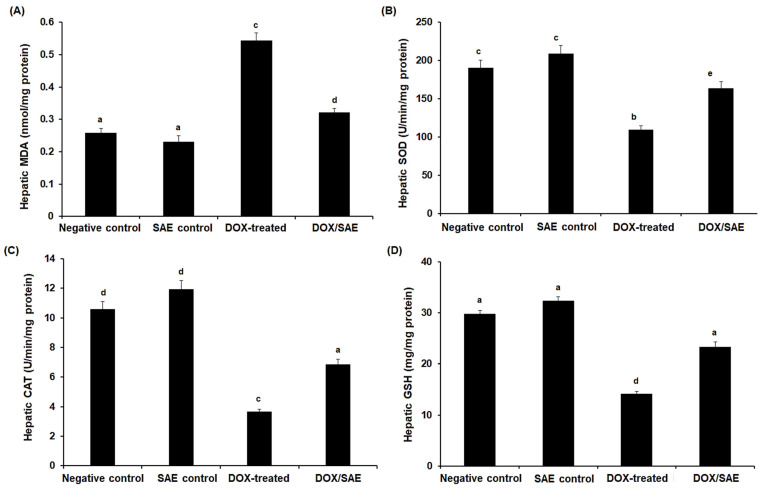
Hepatic malondialdehyde (MDA) (**A**), superoxide dismutase (SOD) (**B**), catalase (CAT) (**C**), and reduced glutathione (GSH) (**D**) levels in the different groups. SAE, *Syzygium aromaticum* extract; DOX, Doxorubicin. The values represent means ± S.D., *n* = 10. Means that do not share a letter showed a significant difference (*p* < 0.05).

**Figure 8 ijms-25-12541-f008:**
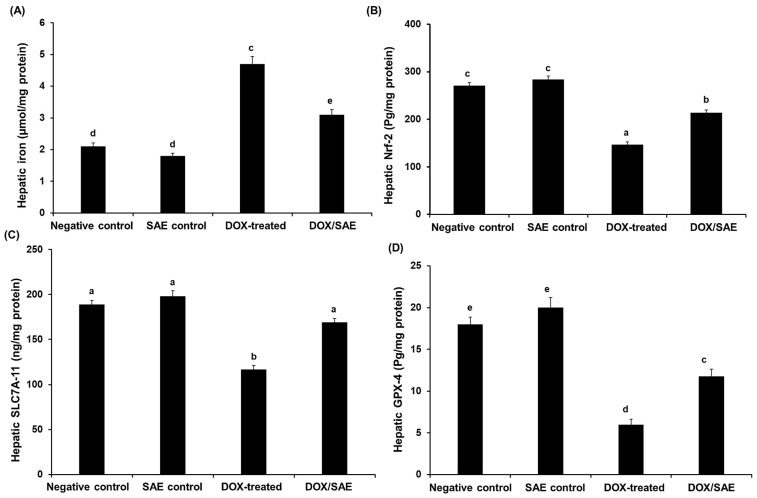
Hepatic iron (**A**), nuclear factor erythroid 2-related factor 2 (NRF-2) (**B**), solute carrier family 7, member 11 (SLC7A-11) (**C**), and glutathione peroxidase (GPX-4) (**D**) in the different groups. SAE, *Syzygium aromaticum* extract; DOX, Doxorubicin. The values represent means ± S.D., *n* = 10. Means that do not share a letter showed a significant difference (*p* < 0.01).

**Figure 9 ijms-25-12541-f009:**
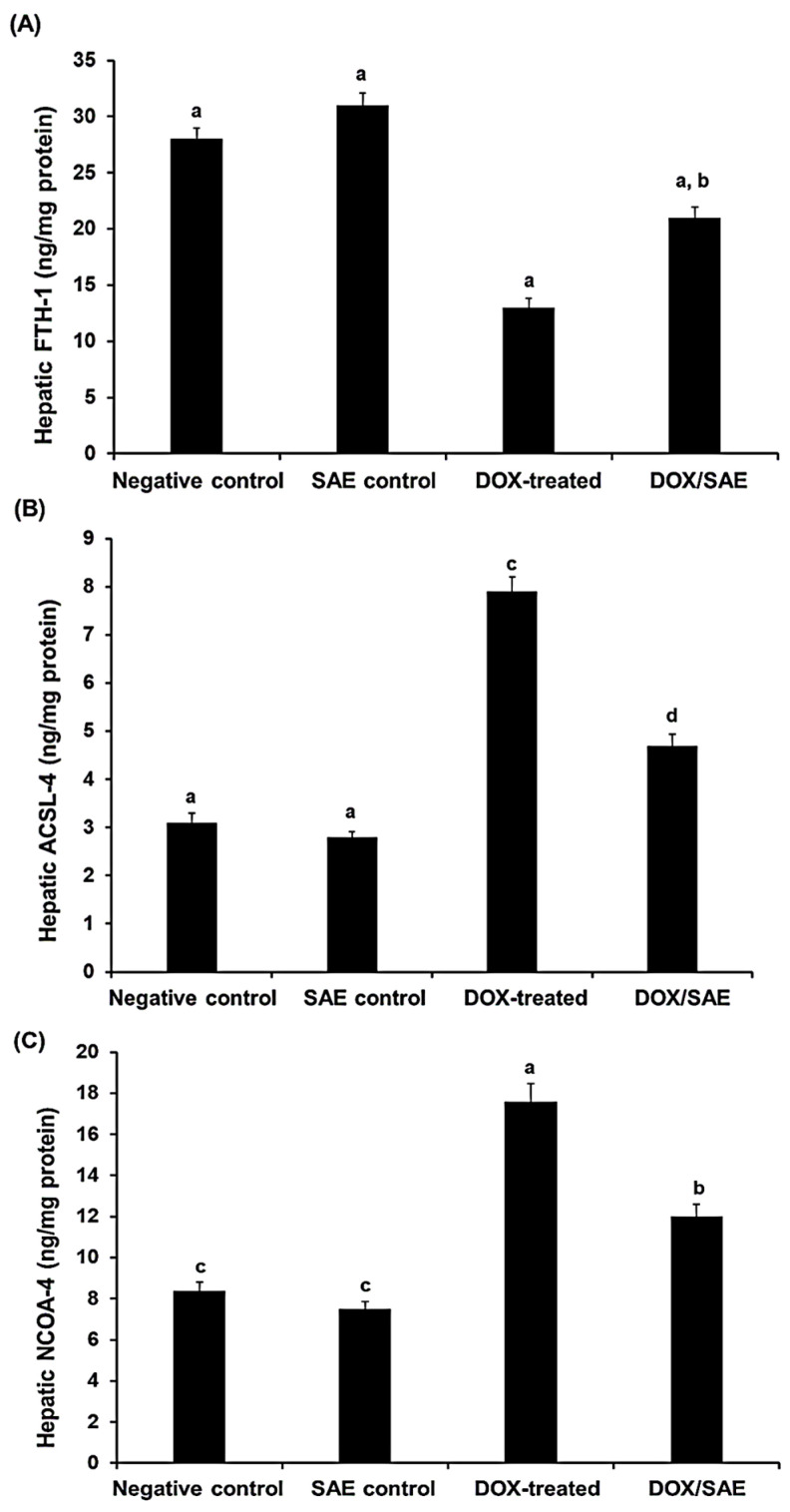
Hepatic ferritin heavy chain-1 (FTH-1) (**A**), acyl–CoA synthetase long-chain family member 4 (ACSL-4), (**B**), and nuclear receptor coactivator 4 (NCOA-4) (**C**) in different groups. The values represent means ± S.D. (*n* = 10). SAE, *Syzygium aromaticum* extract; DOX, Doxorubicin. Means that do not share a letter showed a significant difference (*p* < 0.05).

**Figure 10 ijms-25-12541-f010:**
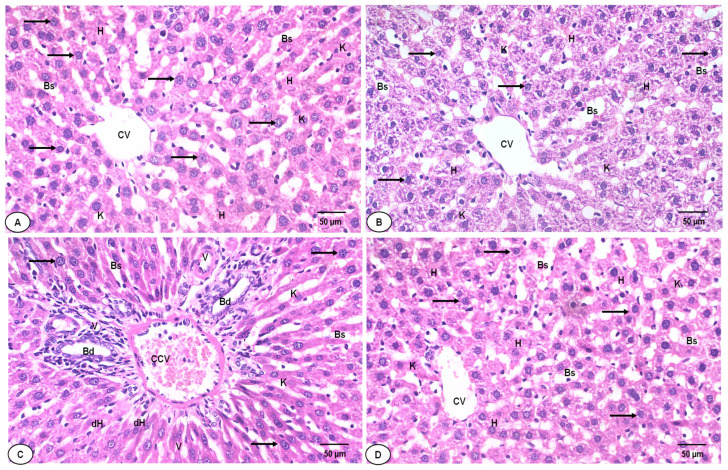
(**A**) A photomicrograph of the liver section of the normal control group shows a normal hepatic structure, regular central veins (CVs), normal hepatocytes (Hs), normal blood sinusoids (Bs), and Kupffer cells (Ks). (**B**) Liver section of the SAE control group shows mostly normal hepatocytes, with normal Bs and Ks. (**C**) Liver section of the DOX-injected group exhibits disorganization of the hepatic architecture (dH), congested CVs (CCVs), cellular infiltrations, a vacuolated cytoplasm (V), pyknotic nuclei (arrows), irregular blood sinusoids (Bds), and distinct Ks. (**D**) Liver section of the DOX–SAE-treated group shows an improvement in the hepatic organization, represented by fewer congestions in the central vein (CV), fewer binucleated hepatocytes, and fewer cellular infiltrations. SAE, *Syzygium aromaticum* extract; DOX, Doxorubicin. (H&E × 400, scale bar = 50 μm).

**Table 1 ijms-25-12541-t001:** Quantitative phytochemical analysis of *S. aromaticum* flower buds (SAFBs).

Phytochemical Analysis	SAFB
Total phenolic content (mg GAE/g DW)	31.86 ± 2.43
Total flavonoid content (mg QE/g DW)	18.65 ± 1.58
Total antioxidant capacity (TAC) (mg AAE/g DW)	69.37 ± 3.79
Saponin (mg/g DW)	365 ± 4.37
Anthocyanin (mg ECG/g DW)	7.89 ± 0.45
DPPH scavenging activity (%)	82.64% ± 3.95
IC_50_ of DPPH (mg/mL)	6.05 ± 0.86

SAFB, *Syzygium aromaticum* flower bud; GAE, Gallic acid equivalent; QE, Quercetin equivalents; DW, Dry weight; TAC, Total antioxidant capacity; AAE, Ascorbic acid equivalent; ECG, Epicatechin gallate; DPPH, Diphenyl-1-picrylhydrazyl; IC_50_, Half-maximal inhibitory concentration.

**Table 2 ijms-25-12541-t002:** GC-MS analysis of the phytochemical composition in SAE.

No.	RT (min)	Name	MF.	P.A (%)
1	7.64	Ethyl-α-D-glucopyranoside	C_8_H_16_O_6_	2.27
2	12.25	Eugenol	C_10_H_12_O_2_	64.17
3	15.82	Caryophyllene	C_15_H_24_	18.07
4	16.54	Humulene	C_15_H_24_	3.83
5	19.50	Caryophyllene oxide	C_15_H_24_O	0.86
6	21.55	10-Heptadecen-8-ynoic acid, methyl ester	C_15_H_24_O	0.67

SAE, *Syzygium aromaticum* extract; RT, Retention time; MF, Molecular formula; P.A%, Peak area percentage.

**Table 3 ijms-25-12541-t003:** ADMET screening for each of DOX and Eugenol.

Compound (A1:S4)	DOX	Eugenol	10-Heptadecen-8-ynoic Acid, Methyl Ester
LogS	−2.26	−2.286	−5.812
LogD	0.473	2.418	4.205
LogP	1.375	2.291	5.78
Pgp-inh	0.001	0.003	0.389
Pgp-sub	0.998	0	0.002
HIA	0.829	0.007	0.004
F (20%)	0.055	0.732	0.208
F (30%)	0.209	0.967	0.975
Caco-2	−6.167	−4.373	−4.457
MDCK	6.34 × 10^−6^	3.01 × 10^−5^	2.52 × 10^−5^
BBB	0.015	0.188	0.713
PPB	90.86%	92.12%	98.58%
VDss	1.177	0.833	1.36
Fu	10.82%	3.22%	0.83%
CYP1A2-inh	0.274	0.901	0.924
CYP1A2-sub	0.495	0.941	0.558
CYP2C19-inh	0.012	0.716	0.882
CYP2C19-sub	0.064	0.659	0.542
CYP2C9-inh	0.006	0.313	0.708
CYP2C9-sub	0.406	0.875	0.992
CYP2D6-inh	0.003	0.85	0.633
CYP2D6-sub	0.193	0.921	0.641
CYP3A4-inh	0.088	0.288	0.858
CYP3A4-sub	0.153	0.371	0.129
CL	13.025	14.042	4.467
T1/2	0.73	0.887	0.379
hERG	0.025	0.017	0.098
H-HT	0.261	0.036	0.561
DILI	0.974	0.046	0.91
Ames	0.829	0.066	0.111
ROA	0.041	0.121	0.026
FDAMDD	0.323	0.153	0.024
Carcinogenicity	0.68	0.414	0.267
EC	0.003	0.713	0.933
EI	0.012	0.982	0.95
Respiratory	0.891	0.51	0.943
BCF	0.567	1.203	2.424
IGC_50_	3.734	3.642	5.204
LC_50_	3.653	3.926	5.874
LC_50_DM	5.55	4.763	5.836
NR-AR	0.026	0.089	0.363
NR-AR-LBD	0.793	0.022	0.005
NR-AhR	0.866	0.413	0.007
NR-Aromatase	0.644	0.02	0.442
NR-ER	0.217	0.224	0.191
NR-ER-LBD	0.588	0.036	0.016
NR-PPAR-gamma	0.221	0.024	0.593
SR-ARE	0.771	0.14	0.336
SR-ATAD5	0.642	0.238	0.022
SR-HSE	0.027	0.094	0.216
SR-MMP	0.945	0.155	0.012
SR-p53	0.979	0.033	0.124
MW	543.17	164.08	278.22
Vol	516.727	179.994	326.919
Dense	1.051	0.912	0.851
nHA	12	2	2
nHD	9	1	0
TPSA	212.39	29.46	26.3
nRot	5	3	12
nRing	5	1	0
MaxRing	18	6	0
nHet	12	2	2
nRig	28	7	0
Flex	0.179	0.429	4
nStereo	5	0	0
Genotoxic Carcinogenicity Mutagenicity	4	0	0
SureChEMBL	1	0	1
Non-Biodegradable	3	0	0
Skin Sensitization	8	5	0
Toxicophores	3	2	2
QED	0.147	0.693	0.301
Synth	5.015	1.961	2.503
Fsp3	0.37	0.2	0.722
MCE-18	118.243	6	0
Natural-Product-likeness	1.37	1.053	1.653
Lipinski	Rejected	Accepted	Accepted
Pfizer	Accepted	Accepted	Rejected
GSK	Rejected	Accepted	Accepted

LogS, Log of the aqueous solubility; LogD, Log at physiological pH (7.4); LogP, Log of the octanol–water partition coefficient; Pgp-inh, P-glycoprotein inhibitors; Pgp-sub, P-glycoprotein substrates; HIA, human intestinal absorption; F (%), oral bioavailability; MDCK, Madin–Darby canine kidney; BBB, blood–brain barrier; PPB, plasma protein binding; VDss, volume of distribution; Fu, fraction unbound; CYP1A1, cytochrome P450 family 1 subfamily A; CL, clearance; T1/2, half-life time; hERG, human ether-à-go-go-related gene; H-HT, hereditary hemorrhagic telangiectasia; DILI, drug-induced liver injury; ROA, route of administration; FDAMDD, Food and Drug Administration maximum daily dose; EC, enteric coated; EI, enzyme induction; BCF, bioconcentration factor; IGC_50_, median inhibitory growth concentration; LC_50_, median lethal concentration; NR-AR-LBD, nuclear receptor–androgen receptor–ligand binding domain; AhR, aryl hydrocarbon receptor; ER, extended-release; SR, sustained-release; ATAD5, ATPase family A domain-containing protein 5; HSE, health, safety, and environment; MMP, matrix metalloproteinase inhibitor; MW, molecular weight; nHA, number of hydrogen bond acceptors; nHD, number of hydrogen bond donors; TPSA, topological polar surface area; nROT, number of rotatable bonds; nHet, number of heteroatoms; MaxRing, number of atoms in the biggest ring; nRing, number of rings; Flex, flexibility; nStereo, number of stereocenters; QED, quantitative estimate of drug-likeness; Fsp3, fraction of carbon atoms that are sp3 hybridized; MCE-18, medicinal chemistry evolution, 2018; GSK, GlaxoSmithKline.

**Table 4 ijms-25-12541-t004:** The ∆G and binding affinity (Kcal/mol).

Compound	GPX-4	NRF-2	SLC7A-11
Doxorubicin	−6.7	−7.8	−8.4
10-Heptadecen-8-ynoic acid, methyl ester	−3.9	−4.3	−5.3
Caryophyllene	−5.2	−6.1	−7.3
Caryophyllene oxide	−5.2	−5.9	−7.3
Ethyl_α_D-glucopyranoside	−4.5	−5.2	−6.6
Eugenol	−4.3	−4.7	−6.0
Humulene	−5.3	−5.9	−7.4

**Table 5 ijms-25-12541-t005:** Serum alanine transaminase (ALT), aspartate transaminase (AST), alkaline phosphatase (ALP), total bilirubin (T.B), and direct bilirubin (D.B) in different groups.

Groups	ALT (U/L)	AST (U/L)	ALP (U/L)	T.B. (mg/dL)	D.B. (mg/dL)
Negative control	25.87 ± 0.76 ^e^	36.27 ± 1.21 ^c^	265.32 ± 4.67 ^a^	0.64 ± 0.013 ^a^	0.132 ± 0.004 ^c^
SAE control	23.44 ± 0.65 ^e^	32.44 ± 1.15 ^c^	258.79 ± 5.23 ^a^	0.58 ± 0.012 ^a^	0.126 ± 0.007 ^c^
DOX-treated	71.23 ± 1.78 ^a^	103.67 ± 2.78 ^e^	486.98 ± 6.34 ^b^	1.45 ± 0.034 ^b^	0.412 ± 0.008 ^a^
DOX–SAE	38.93 ± 0.85 ^d,e^	60.78 ± 2.56 ^d^	317.92 ± 5.79 ^e^	0.90 ± 0.016 ^a,c^	0.215 ± 0.006 ^d^

The values represent the mean ± S.D., *n* = 10. SAE, *Syzygium aromaticum* extract; DOX, Doxorubicin. Means that do not share a letter in each column showed a significant difference (*p* < 0.05).

**Table 6 ijms-25-12541-t006:** The relative mRNA expression levels of the *NRF2*, *SLC7A11*, *GPX4*, *FTH1*, and *NCOA4* genes of the different groups.

Groups	*NRF2*	*SLC7A11*	*GPX4*	*FTH1*	*NCOA4*
Negative control	1.00 ± 0.00 ^c^	1.00 ± 0.00 ^a^	1.00 ± 0.00 ^e^	1.00 ± 0.00 ^b^	1.00 ± 0.00 ^a^
SAE control	1.56 ± 0.09 ^f^	1.12 ± 0.07 ^a^	1.39 ± 0.06 ^c^	1.07 ± 0.03 ^b^	1.02 ± 0.009 ^a^
DOX-treated	0.39 ± 0.04 ^a^	0.62 ± 0.08 ^b^	0.70 ± 0.05 ^b^	0.25 ± 0.06 ^a^	3.67 ± 0.12 ^b^
DOX–SAE	0.89 ± 0.07 ^c^	0.78 ± 0.06 ^a,b^	0.95 ± 0.08 ^e^	0.61 ± 0.08 ^c^	1.57 ± 0.08 ^a,c^

The values represent the mean ± S.D., *n* = 10. SAE, *Syzygium aromaticum* extract; DOX, Doxorubicin. Means that do not share a letter in each column show a significant difference (*p* < 0.001).

**Table 7 ijms-25-12541-t007:** Hepatic inflammatory cytokines, including interleukin-6 (IL-6), interleukin-1β (IL-1β), tumor necrosis factor-α (TNF-α), nuclear factor kappa-B (NF-κB), and cyclooxygenase-2 (COX-2), in the different groups.

Groups	IL-6 (Pg/mg Tissue)	IL-1β (Pg/mg Tissue)	TNF-α (Pg/mg Tissue)	NF-κB (Pg/mg Tissue)	COX-2 (Pg/mg Tissue)
Negative control	6.83 ± 0.56 ^e^	11.86 ± 1.15 ^a^	3.85 ± 0.29 ^c^	185.61 ± 4.23 ^a^	317.82 ± 6.48 ^c^
SAE control	6.12 ± 0.65 ^e^	12.08 ± 0.87 ^a^	3.27 ± 0.32 ^c^	170.85 ± 4.63 ^a^	296.73 ± 5.86 ^c^
DOX-treated	19.47 ± 0.93 ^a^	29.79 ± 1.85 ^c^	11.21 ± 1.04 ^b^	385.43 ± 6.24 ^b^	587.92 ± 7.58 ^a^
DOX–SAE	9.87 ± 0.76 ^d,e^	18.45 ± 1.06 ^e^	6.79 ± 0.75 ^e^	245.92 ± 5.85 ^c^	402.21 ± 6.46 ^d^

The values represent the mean ± S.D., *n* = 10. SAE, *Syzygium aromaticum* extract; DOX, Doxorubicin. Means that do not share a letter in each column show significant differences (*p* < 0.001) or (*p* < 0.01).

**Table 8 ijms-25-12541-t008:** Effect of SAE treatment on the liver histopathologic score of the different groups.

Groups	Histopathologic Score
Negative control	0.12 ± 0.09 ^a^
SAE control	0.10 ± 0.11 ^a^
DOX-treated	3.50 ± 0.21 ^c^
DOX–SAE	1.82 ± 0.19 ^b^

Means that do not share a letter in each column show significant differences (*p* < 0.01).

**Table 9 ijms-25-12541-t009:** Forward and reverse primer sequences for RT-PCR.

Gene	Accession Number	Forward Sequence (5′–3′)	Reverse Sequence (5′–3′)
*NRF2*	NM_031789.3	CACATCCAGACAGACACCAGT	CTACAAATGGGAATGTCTCTGC
*SLC7A11*	NM_001107673.3	GAGGCGCTGTAGCCACATTA	GGCATTCAACCAGGTGATCC
*GPX4*	NM_008162	CTCCATGCACGAATTCTCAG	ACGTCAGTTTTGCCTCATTG
*FTH1*	NM_012848.2	CCCTTTGCAACTTCGTCGCT	CTCCGAGTCCTGGTGGTAGT
*NCOA4*	NM_001034007.1	TGAAGTGCAGTGCTCACACA	TTCGCTGCTGCTGACAGTTA
*GAPDH*	NM_017008.4	CCGCATCTTCTTGTGCAGTG	GAGAAGGCAGCCCTGGTAAC

*NRF2*, Nuclear factor erythroid 2-related factor 2; *SLC7A11*, Solute carrier family 7, member 11; *GPX4*, Glutathione peroxidase; *FTH1*, Ferritin heavy chain-1; *NCOA4*, Nuclear receptor co-activator-4; *GAPDH*, Glyceraldehyde-3-phosphate dehydrogenase.

## Data Availability

The data presented in this study are available on request from the corresponding author. The data are not publicly available due to ethical restrictions, our research still be used in ongoing studies, future research, and sharing it prematurely could jeopardize the intellectual property or future publications.

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
