# Peer review of "Syzygium aromaticum Extract Mitigates Doxorubicin-Induced Hepatotoxicity in Male Rats"

_ijms, 2024, doi:10.3390/ijms252312541_

Round 1
Reviewer 1 Report (New Reviewer)
Comments and Suggestions for Authors In this study, authors investigated the effect of S. aromaticum extract (SAE) on hepatotoxicity promoted by DOX in rats using insilico, biochemichal and histological analysis. - I do see that the major concern is the gaps between the computational analysis and the in vitro as well as the in vivo work. If the authors managed to show the specific interaction with important signaling and the highlited compounds, logically an in vitro evedience should be shown such as western blot. That may not be required if it is pure bioinformatics study. - Pre-docking energy minimization steps are not clear, please descibe how those were done and provide a supplementary data shown the interaction with and without energy minimization. - Are these binding sites are specific and important for inducing inhibition, or just docking results were analyzed with out relate them to the inhibition of the active residues required for those molecules activities. - In the discussion, 3rd paragraph, line 348-388, no reference is cited, again please clarify the docking was done to show a specific interactions and then inhibition, not just docking for docking. - At this stage, the study is not acceptable, and the above has to be done. Comments on the Quality of English Languageminor cqan be controlled in the next version
Author Response
Dear Reviewer, # 1,
Thank you for the review of our manuscript (ijms-3199965) titled “Syzygium aromaticum extract mitigates doxorubicin-induced hepatotoxicity via targeting ferroptosis and NRF-2/SLC7A-11/GPX-4 pathway in male rats”. The whole manuscript has been revised carefully and we have considered all your comments. These revisions are highlighted in the manuscript by track changes and are summarized in the attached file.

Reviewer 2 Report (Previous Reviewer 2)
Comments and Suggestions for Authors I read with attention re rebuttal by the authors. They indeed made their point but my major points remain unsolved. I also rapidly read the manuscript and I could not easily find and significant different in comparison with the previous one as the possible changes were not clearly marked and made visible. Therefore, my points remain untouched.Author Response
Dear Reviewer, # 2,
Thank you for the review of our manuscript (ijms-3199965) titled “Syzygium aromaticum extract mitigates doxorubicin-induced hepatotoxicity via targeting ferroptosis and NRF 2/SLC7A-11/GPX-4 pathway in male rats”.
I also rapidly read the manuscript, and I could not easily find, and significant different in comparison with the previous one as the possible changes were not clearly marked and made visible.
In the revised manuscript, we made marked and visible changes regarding our in silico studies, and the whole manuscript was revised carefully with significant changes from the first submission. In our novel study, for the first time, we investigated the ameliorative effect of SAE on DOX-induced hepatotoxicity and addressed its mechanism via targeting oxidative stress, inflammation, ferroptosis, and NRF 2/SLC7A-11/GPX-4 pathway in rats.
We would like to thank you for helping to increase the quality of this paper. We hope the changes that have been made were appropriate and the manuscript can now be accepted for publication. Please address all correspondence to the author indicated below.
Sincerely Yours,
Corresponding author
Prof. Dr. Karim Samy El-Said

Reviewer 3 Report (New Reviewer)
Comments and Suggestions for Authors
Manuscript ID:ijms-3272211
The manuscript by Alsirhani A. M. et al. presents SAE, a kind of Syzygium aromaticum extract, can mitigate doxorubicin-induced hepatotoxicity via targeting ferroptosis and NRF-2/SLC7A-11/GPX-4 pathway in male rats, it is interesting work that it may provide a protective agent SAE to alleviate hepatotoxicity induced by doxorubicin, and SAE co-treatment with doxorubicin is further linked to the upregulation of Nrf-2, the inhibition of oxidative stress and ferroptosis, in comparison to doxorubicin used alone. However, the manuscript has obvious drawbacks, I could not recommend it to be accepted for publication in its present form in this journal.
Comments and suggestion:
1. The title of the manuscript: lack of evidence on the crucial genes to be knocked out or down, the authors cannot claim that Syzygium aromaticum extract can target ferroptosis or NRF-2/SLC7A-11/GPX-4 pathway. Their data presented only support that there is a positive correlation between the SAE treatment and NRF-2/SLC7A-11/GPX-4 pathway and related response.
2. combined 2.1 and 2.2.
3. All the abbreviations in Figure2 should be shown in fully form in legend, and all the abbreviations in the first column in Table3 should be shown in fully form. The results in Figure2 and Table3 should be described in more detailed in In Silico ADMET Analysis.
4.combined 2.5 and 2.6, and deleted Figure6. Check the Figure6A cited on line193, is it Figure7A? Changed Figure7A into relative body weight on vertical axis, check the values of Figure7B, why do the relative liver weights only reach less than 5?
5. The statistically significance at a level of P or p <0.05? It presents in two different forms in Line 190 and 192, it should be typed in Italic style in the whole manuscript.
6. Changed the title of 2.9, the result only showed The effect of SAE on ferroptosis and NRF-2/SLC7A-11/GPX-4 pathway.
7. Figure9 and 10, parts of them are missing.
8.combined 2.9 and 2.10, and moved Table6 into Support information together with 4.9 section, the authors should address how did they measure protein levels in Figure9 and Figure10. Delete Figure10A.
9. delete each in the note of Table 8 on Line301, there is only one column. Give full name of all the abbreviations in Figure11.
10. The experiment design lacks different dosage of SAE, whether the responses are in dose-dependent manner has not been shown, and no information on the dose ranges of SAE and its effective responses.
11.The results referred in discussion should be cited.
12. Why only male rats were chosen to be used in the present study should be stated in Method section or other appropriate part.
13. The discussion section is lengthy, it should be simplified as far as possible. Especially, delete Line339-346.
Rewrite Line415 which confirmed the confirmed the clove hepatoprotective action of clove;
Delete Line415-416 Therefore, it's plausible that SAE hepatoprotective mechanism results from its antioxidant activity. Or, the authors should measure the antioxidant activities of SAE.
The authors claims on Line434-435 SAE could mitigate liver injury by inhibiting liver ferroptosis through activating the SLC7A-11/GPX-4 pathway in DOX-administered rats. I suggested the sentence should be deleted, the results have not given any direct evidence.
Rewrite Line463 Enzymes called cytoplasmic reductase and cytochrome.
14. Conclusion: rewrite Line573-575 and can be used as a new potent ferroptosis inhibitor in the liver tissues of rats injected with DOX. The combination of SAE and DOX could serve as a new strategy for efficient chemotherapy. The sentence seems over claimed the results.
Author Response
Dear Reviewer, # 3,
Thank you for the review of our manuscript (ijms-3199965) titled “Syzygium aromaticum extract mitigates doxorubicin-induced hepatotoxicity via targeting ferroptosis and NRF 2/SLC7A-11/GPX-4 pathway in male rats”. The whole manuscript has been revised carefully and we have considered all your comments. These revisions are highlighted in the manuscript by track changes and are summarized in the attached file.

Round 2
Reviewer 1 Report (New Reviewer)
Comments and Suggestions for Authors
In their response to the previous comments, authors selected the comments which they can respond to and ignored others, so please answer the following comments:
Comment 1: Authors did not answer my question about the gap betwwen the in sillico and the preclinical work (the in vitro and in vivo work)? Since they are going to do in vitro and in vivo work why did not they do the in vito interaction beteen the highlited compounds and at least any of NRF-2, SLC7A-11, and 134 GPX-4-- ex: through immunoprecepitation assay.
Comment 2: Authors did not answer my inquiry about predocking energy minimization steps. Simply unsuccessful minimization is usually affecting the docking energy results where an unknown stereoisomer, or a high energy conformer, can lead to high are meaningless. binding interactions. So to convice the reviewer, as requested before, please submit a supplementary file shows just example of the docking results with and without energy minimization step (final discovery study docking energy result of both).
Comment 3: Do the protein structures retrived from Uniprot NRF-2, SLC7A-11, and 134 GPX-4 have comparable codes at the protein database?
Comment 4: I hvae asked about the specific binding to the amino acids residues, seeking for results interpretation about the inhibitroy effect of your extract highlited compounds and the major signialing molecules such as NRF-2, but got no answer. For example as in Figure 5, (A) The interaction with NRF-2, the conventional H-bond 175 (LYS553): Is LYS553 is important for NRF2 activity or function or any other docking data? References are needed.
Comment 5: In the previous version I have written ' In the discussion, 3rd paragraph, lines 348-388, no reference is cited, again please clarify the docking was done to show a specific interactions and then inhibition, not just docking for docking'--
The authors responded 'In the revised manuscript, the 3rd paragraph, lines from 348-388, were cited with references.'
Actually, no refernce has been added in this paragraph in the currently revised version, that is because authors need to think in the effcet of the specific binging and its inhibitory effect on the functional part of the protein (this part should be required for the protein function and/or activation).
I consider that the docking should serve the preclinical work and without carful revision I may not recommend acceptance in the future.
Author Response
Dear Reviewer, # 1,
Thank you for the review of our manuscript (ijms-3199965) titled “Syzygium aromaticum extract mitigates doxorubicin-induced hepatotoxicity via targeting ferroptosis and NRF-2/SLC7A-11/GPX-4 pathway in male rats”. The whole manuscript has been revised carefully and we have considered all your comments. These revisions are highlighted in the manuscript by track changes

Reviewer 3 Report (New Reviewer)
Comments and Suggestions for Authors
Manuscript ID:ijms-3272211v2
I had to reject this manucript for publication in this Journal, because the manuscript has not been improved by the authors at all than the original version, except that there is only a little change in the title.
Comments on the Quality of English LanguageNone.
Author Response
Dear Reviewer, # 3,
Thank you for the review of our manuscript (ijms-3199965) titled “Syzygium aromaticum extract mitigates doxorubicin-induced hepatotoxicity via targeting ferroptosis and NRF 2/SLC7A-11/GPX-4 pathway in male rats”. The whole manuscript has been revised carefully and we have considered all your comments. These revisions are highlighted in the manuscript by track changes

Round 3
Reviewer 1 Report (New Reviewer)
Comments and Suggestions for Authors
Authors did effort to answer most of the comments
Reviewer 3 Report (New Reviewer)
Comments and Suggestions for Authors
Manuscript ID:ijms-3272211v3
All the question has been resolved, and I recommend it to be accepted for publication in this Journal.
This manuscript is a resubmission of an earlier submission. The following is a list of the peer review reports and author responses from that submission.
Round 1
Reviewer 1 Report
Comments and Suggestions for Authors
This article investigates the hepatoprotective effects of Syzygium aromaticum extract (SAE) against doxorubicin (DOX)-induced hepatotoxicity in male rats, focusing on the inhibition of ferroptosis through the modulation of the NRF-2/SLC7A-11/GPX-4 signaling pathway. The study provides valuable insights into the potential therapeutic use of SAE in mitigating DOX-induced liver damage. Therefore, I suggest it be accepted after the author complements the content, articulates the obscure content, and corrects the format.
1. It is suggested that ‘DOX-intoxicated group’ be amended to ‘DOX-exposed group’, as the term ‘intoxicated’ may be misinterpreted. The word ‘intoxicated’ can be misunderstood. In a scientific context, it would be clearer to use ‘exposed’. 2.
2. There is inconsistency in the way Nrf2 is written in the text, and it is recommended to ensure that ‘NRF-2’ is consistently referred to as ‘Nrf2’ throughout the text.
3. line 283, Histopathological description, suggests more precise terminology, ‘a significant improvement in the hepatic structure and less congestion’ suggests ‘a significant improvement in hepatic architecture and less congestion’. In line 283, a more precise terminology is suggested for the histopathological description.
4. In line 369, ‘These interaction patterns’ is a plural subject, but the subsequent predicate structure uses the singular form. It is suggested that ‘DOX's binding is’ be changed to It is suggested that ‘DOX's bindings are’ be replaced by ‘DOX's bindings are’, or that the sentence structure be adjusted to avoid subject-predicate inconsistency.
5. the format of references is not uniform, some of the references lack complete DOI links, and some of the formats do not meet the requirements of the journals.
6. the redundant symbol ‘...’ appears in line 233. .
7. there are several inconsistencies between the numbering of figures and the citations in the text, it is recommended to double-check all the figure numbers and the citations in the text. For example, in row 219, ‘Figure 4A’ needs to be changed to ‘Figure 8A’.
8. Inconsistent alignment of data in Table 4. Ensure that the alignment of data in all tables is consistent, especially when comparing different experimental groups.
9. lines 81-82 ‘Moreover, a previous study reported the efficacy of the compound contained in beetroot ethanol extract in ameliorating hepatic damage induced by DOX in rats.’ This expression is ambiguous and does not detail the specific compounds mentioned and the results of the study.
Author Response
Dear Reviewer, # 1,
Thank you for the review of our manuscript (ijms-3199965) titled “Syzygium aromaticum extract mitigates doxorubicin-induced hepatotoxicity via targeting ferroptosis and NRF 2/SLC7A-11/GPX-4 pathway in male rats”. The whole manuscript has been revised carefully and we have considered all your comments. These revisions are highlighted in the manuscript by track changes.

Reviewer 2 Report
Comments and Suggestions for Authors
The paper by Alsirhani and coworkers deals with a putative protective effect of an extract from S. aromaticum (SAE) on doxorubicin induced hepatotoxicity in male rats.
The study considers a large spectrum of analysis ranging from molecular docking to microscope histological examination.
Authors have also performed an accurate analysis of S. aromaticum by GC-MS reporting that the major component of the extract is eugenol, which is not a surprise considering that this plant belongs to the mitraceae, a family comprising many differnt plants that have been considered for their putative beneficial effects to human (and animal) health.
Based on eugenol abundance in the extract, authors only focused on this molecule when performing the molecular docking analysis and ADMET, excluding a priori that other compounds, less abundant, may have stronger binding properties toward selected targets. This choice actually significantly reduces the importance of this study as many papers have been already published dealing with the biological properties of eugenol (and also about the protective effects of other plants belonging to the myrtaceae family).
The study reports that 340 mg/kg BW administered by gavage (corresponding to about 24 grams for an adult human being) significantly ameliorate the majority of detrimental/harmful effects of DOX administration to the liver of healthy animals. SAE dose is indeed very high and even considering the different metabolic characterisitics of rodents, is unphysiological and not achievable in humans. Therefore, data obtained and conclusion cannot be referred to humans.
There are other "minor" points that this referee is not specifically mentioning here, considering that the points above strongly reduce the scientific significance of data reported in this manuscript (e.g. is the SAE composition constant in differnt batches? why only males have been considered? estrogens may have an important impact on SAE and DOX effects. Results are not sufficiently discussed and the discussion section is somehow unbound of the data obtained and no critical points are identified.)
Overall, this study has some merits but it is neither very original nor informative. I doubt that the weaknesses identified can be solved by additional experiments or even just by an editorial shaping following a point-to-point rebuttal.
Author Response
Dear Reviewer, # 2,
Thank you for the review of our manuscript (ijms-3199965) titled “Syzygium aromaticum extract mitigates doxorubicin-induced hepatotoxicity via targeting ferroptosis and NRF 2/SLC7A-11/GPX-4 pathway in male rats”. The whole manuscript has been revised carefully and we have considered all your comments. These revisions are highlighted in the manuscript by track changes.
We thank you for helping to increase the quality of this paper. We hope the changes that have been made were appropriate and the manuscript can now be accepted for publication. Please address all correspondence to the author indicated below.
Sincerely Yours,
Corresponding author
Prof. Dr. Karim Samy El-Said

Round 2
Reviewer 2 Report
Comments and Suggestions for Authors
Authors have nicely replied to my comments but, in my opinion, they did not solve the flaws identified after the first submission. The novelty of this study is very very scarce.